# ATAD2 controls chromatin-bound HIRA turnover

Tao Wang[1,*], Daniel Perazza[1,*], Fayçal Boussouar[1], Matteo Cattaneo[1], Alexandre Bougdour[1], Florent Chuffart[1], Sophie Barral[1], Alexandra Vargas[1], Ariadni Liakopoulou[1], Denis Puthier[2], Lisa Bargier[2], Yuichi Morozumi[1,3], Mahya Jamshidikia[1], Isabel Garcia-Saez[4], Carlo Petosa[4], Sophie Rousseaux[1], André Verdel[1], Saadi Khochbin[1]

**Taking advantage of the evolutionary conserved nature of ATAD2, we report here a series of parallel functional studies in human, mouse, and *Schizosaccharomyces pombe* to investigate ATAD2's conserved functions. In *S. pombe*, the deletion of *ATAD2* ortholog, *abo1*, leads to a dramatic decrease in cell growth, with the appearance of suppressor clones recovering normal growth. The identification of the corresponding suppressor mutations revealed a strong genetic interaction between Abo1 and the histone chaperone HIRA. In human cancer cell lines and in mouse embryonic stem cells, we observed that the KO of *ATAD2* leads to an accumulation of HIRA. A ChIP-seq mapping of nucleosome-bound HIRA and FACT in *Atad2* KO mouse ES cells demonstrated that both chaperones are trapped on nucleosomes at the transcription start sites of active genes, resulting in the abnormal presence of a chaperone-bound nucleosome on the TSS-associated nucleosome-free regions. Overall, these data highlight an important layer of regulation of chromatin dynamics ensuring the turnover of histone-bound chaperones.**

## Introduction

ATAD2 belongs to the large AAA-ATPase family involved in a variety of cellular functions (Boussouar et al, 2013; Cattaneo et al, 2014; Nayak et al, 2019). ATAD2 also bears a bromodomain whose preferential cognate ligand is a histone H4 tail peptide acetylated at lysine 5 and lysine 12 (Caron et al, 2010; Koo et al, 2016; Morozumi et al, 2016), a signature of neo-synthesized H4. This particular domain clearly ranks ATAD2 among the chromatin binders and suggests a role as a regulator of chromatin structure and functions, and possibly histone deposition. In addition, ATAD2 is one of the most conserved proteins among eukaryotes, being present from yeast to human (Cattaneo et al, 2014). Early studies on ATAD2 suggested that it could be a factor favouring the activity of oncogenic transcription

factors such as MLL, E2F and Myc (Boussouar et al, 2013). *ATAD2* is a gene normally almost exclusively expressed in male germ cells as well as in embryonic stem (ES) cells (Morozumi et al, 2016). Several studies highlighted its nearly systematic de-repression in malignant cells (Caron et al, 2010). Indeed, *ATAD2* is activated in all solid cancers and its high expression is predictive of poor prognosis in many unrelated tumours (Han et al, 2019). Its activation in all cancers prompted investigators to classify the gene among the most frequently activated members of the so-called cancer/testis (C/T) gene family (Caron et al, 2010).

Almost all of our current knowledge regarding the functions of ATAD2 derives from various cancer-related studies, which suggest a general role for ATAD2 as a pro-oncogenic factor stimulating oncogenic transcription factor activity and cell growth (Caron et al, 2010; Han et al, 2019). The only study that examined ATAD2 in its physiological context of expression, in ES cells, demonstrated that it is required for maintaining the hyperdynamic state of chromatin in these cells. Indeed, Atad2 was found mostly associated with active gene clusters along with acetylated nucleosomes as well as with chromatin-associated factors involved in DNA-replication, DNA repair, chromatin remodelling (Morozumi et al, 2016). In addition, the comparison between Atad2-associated factors in mouse ES cells and the interaction partners of its *S. cerevisiae* homologue, Yta7, showed a remarkable similarity between the two interactomes (Morozumi et al, 2016), suggesting a critical function for ATAD2/Yta7, which would be conserved between the fungal and animal kingdoms.

Additional functional data in both *S. cerevisiae* (Yta7) and *Schizosaccharomyces pombe* (Abo1) suggested that the most probable function of ATAD2 could be its capacity to control histone metabolism and turnover (Lombardi et al, 2011, 2015; Cattaneo et al, 2014; Gal et al, 2016; Shahnejat-Bushehri & Ehrenhofer-Murray, 2020). Along this line, some data led to the hypothesis that ATAD2/Yta7/Abo1 itself could be a histone chaperone involved in the eviction or assembly of histones (Cho et al, 2019; Shahnejat-Bushehri & Ehrenhofer-Murray, 2020). In agreement with these data, in its physiological state of expression (ES cells), we previously reported

---

[1]Centre National de la Recherche Scientifique (CNRS), Unite Mixte de Recherche (UMR) 5309/INSERM U1209/Université Grenoble-Alpes/Institute for Advanced Biosciences, La Tronche, France  [2]Aix Marseille Université, INSERM, Theories and Approaches of Genomic Complexity (TAGC), Transcriptomique et Genomique Marseille-Luminy (TGML), Marseille, France  [3]Division of Biological Science, Nara Institute of Science and Technology, Ikoma, Japan  [4]Université Grenoble Alpes/CNRS/CEA, Institut de Biologie Structurale, Grenoble, France

Correspondence: andre.verdel@univ-grenoble-alpes.fr; saadi.khochbin@univ-grenoble-alpes.fr
*Tao Wang and Daniel Perazza contributed equally to this work

that Atad2 is required to maintain a high rate of histone turnover (Morozumi et al, 2016).

Although many studies, especially in yeast, suggest that Atad2 is a regulator of histone deposition/eviction, the precise molecular basis of this activity remains obscure. Here, to understand the molecular link between ATAD2 function and chromatin dynamics and histone turnover, we re-examined our published proteomic data on endogenous Atad2-associated factors in mouse ES cells and identified the FACT histone chaperone complex. Likewise, a parallel examination of the available Yta7 interactome also highlighted FACT as a histone chaperone associated with this factor (Lambert et al, 2010; Morozumi et al, 2016). In addition, a functional relationship between Abo1 and FACT was also reported in *S. pombe*, where Abo1 was found in a complex with FACT (Gal et al, 2016).

Based on these data, we surmised that ATAD2 might be a regulator of histone chaperone activity, which could explain the molecular basis of ATAD2-controlled histone turnover. To test this hypothesis, we set up a parallel study of ATAD2 activity in a series of fundamentally different model systems, namely, *S. pombe*, human cancer cells and mouse ES cells. These studies highlight a previously unknown layer in the control of histone chaperones' activities, especially that of HIRA, and identify ATAD2 as its master regulator.

# Results

## Mutations in subunits of HIRA or CAF1 complexes and in histones H3 or H4 rescue the growth defect of *S. pombe abo1Δ* cells

The Atad2-dependent histone turnover discovered in mouse ES cells (Morozumi et al, 2016) prompted us to exploit the evolutionary conservation of Atad2 (Fig 1A) in the fission yeast *S. pombe*, to learn more about possible molecular mechanisms that could link Atad2's function to histone turnover. The deletion of the *abo1* gene leads to a severe proliferation defect (Fig 1B, compare colony sizes in the magnified images from *wild-type* and *abo1Δ* plates, and Fig 1C). Interestingly, plating *abo1Δ* cells on solid medium revealed that, despite the clonal nature of the original *abo1Δ* cell cultures, some cells diverged over time, giving rise to colonies able to grow at a rate close to that of wild-type colonies (Fig 1B and C). This led us to hypothesize that *abo1Δ* cells could acquire genomic mutations suppressing their growth defect. Accordingly, we set up a screen using massively parallel sequencing to identify the putative genomic mutations (Fig 1D).

A total of five wild-type, six slow-growing *abo1Δ*, and 14 *abo1Δ* suppressor clones (named *abo1Δ_supA* to *N*, Fig S1) were grown separately and their total RNAs were purified and analysed by sequencing (Fig 1D). We searched for single-nucleotide variants (SNVs) within the RNA-seq data (Table S1). Strikingly, in 11 of the 14 suppressor clones analysed, SNV mapping identified genes encoding histone chaperones or histones (Fig 1E). More specifically, six different SNVs mapped to a single gene, *hip3*, which encodes one of the four subunits of the histone chaperone complex HIRA (Figs 1F and S2A and Table 1). Two other SNVs mapped to the *slm9* gene, which encodes another subunit of the HIRA complex. The remaining

three SNVs mapped to *pcf1*, a gene encoding a subunit of the CAF1 histone chaperone complex, and to two of the genes coding for histones H3 or H4, respectively. To verify that the SNVs identified from RNA sequences correspond to genomic mutations, we sequenced the corresponding DNA regions, for the SNV found in either *abo1Δ_supA*, *B*, or *G* cells (Table 1). As expected, DNA sequencing validated the presence of the point mutations in the genome of the respective suppressor clones (Fig S2B). The contribution of the SNVs to the suppression of the *abo1Δ* cell growth defect was then further validated by backcrossing three of the suppressor clones, *abo1Δ_supA*, *C*, and *H*, with the parental wild-type strain to eliminate other SNVs present in their genome and susceptible to contribute to the suppressor phenotype. In particular, these backcrosses eliminated two SNVs mapping to the *esf1* and *SPCC622.11* genes, respectively, that were present in the vast majority of *abo1Δ* suppressor clones (Fig 1E). Importantly, growth assays conducted with backcrossed cells, which kept the *hip3* or *slm9* SNVs but had lost the SNVs in *esf1* and *SPCC622.11* genes, showed that these cells still suppress the growth defect caused by the deletion of *abo1* gene (Fig S3). In total, our suppressor screen identified eight suppressor mutations concentrated in only two genes, both encoding subunits of the HIRA chaperone complex, therefore highlighting a strong positive genetic interaction between Abo1 and HIRA. In addition, this screen revealed genetic interactions of Abo1 with CAF1 as well as with H3 and H4 histones.

From these data, we hypothesized that the function of Abo1 could be to counteract the histone loading activity of the HIRA and, to some extent, CAF1 histone chaperone complexes and that the absence of Abo1 could result in a net increase in histone loading by these chaperones. The deletion of *abo1* would thus lead to these chaperones being "trapped" on chromatin (Fig 1G). The increased residence time of histone chaperones on chromatin could both be toxic and interfere with the growth of *abo1Δ* cells.

## Histones H3 and H4 underdosage rescues *abo1Δ* cell toxicity

The genetic screen for *abo1Δ* suppressors also identified point mutations in the histone H3 and H4 genes (Fig 1F and Table 1). Because both the inactivating mutations of histone chaperones and of histones themselves rescue *abo1Δ*'s cell proliferation defect, we reasoned that histone overloading could be the source of cell toxicity. Following this idea, the prediction is that a targeted reduction of histone dosage in *abo1Δ* cells should rescue the growth defect. To test this hypothesis, we monitored the effect of deleting the H3 and H4 genes in Abo1 deficient cells. To avoid the acquisition of unwanted SNVs in *abo1Δ* cells during the experiment, which could potentially interfere with the effect caused by the deletion of histone genes, we used a N-degron conditional knock-down (KD) system (Fig 2A). This system allowed efficient KD of Abo1 (Fig 2B) and a slow growth defect visible 8 h after the induction of Abo1 KD (Fig S4). In *S. pombe*, the histone H3 and H4 genes are grouped in three H3/H4 gene pairs. We deleted independently two of the three H3/H4 genes pairs, namely, *hht1/hhf1* and *hht2/hhf2*. As expected, the KD of Abo1 on its own led to a marked growth defect. However, the co-deletion of either the *hht1/hhf1* or *hht2/hhf2* gene pair eliminated most of the growth defect caused by Abo1's induced downregulation (Fig 2C). These observations support our hypothesis that

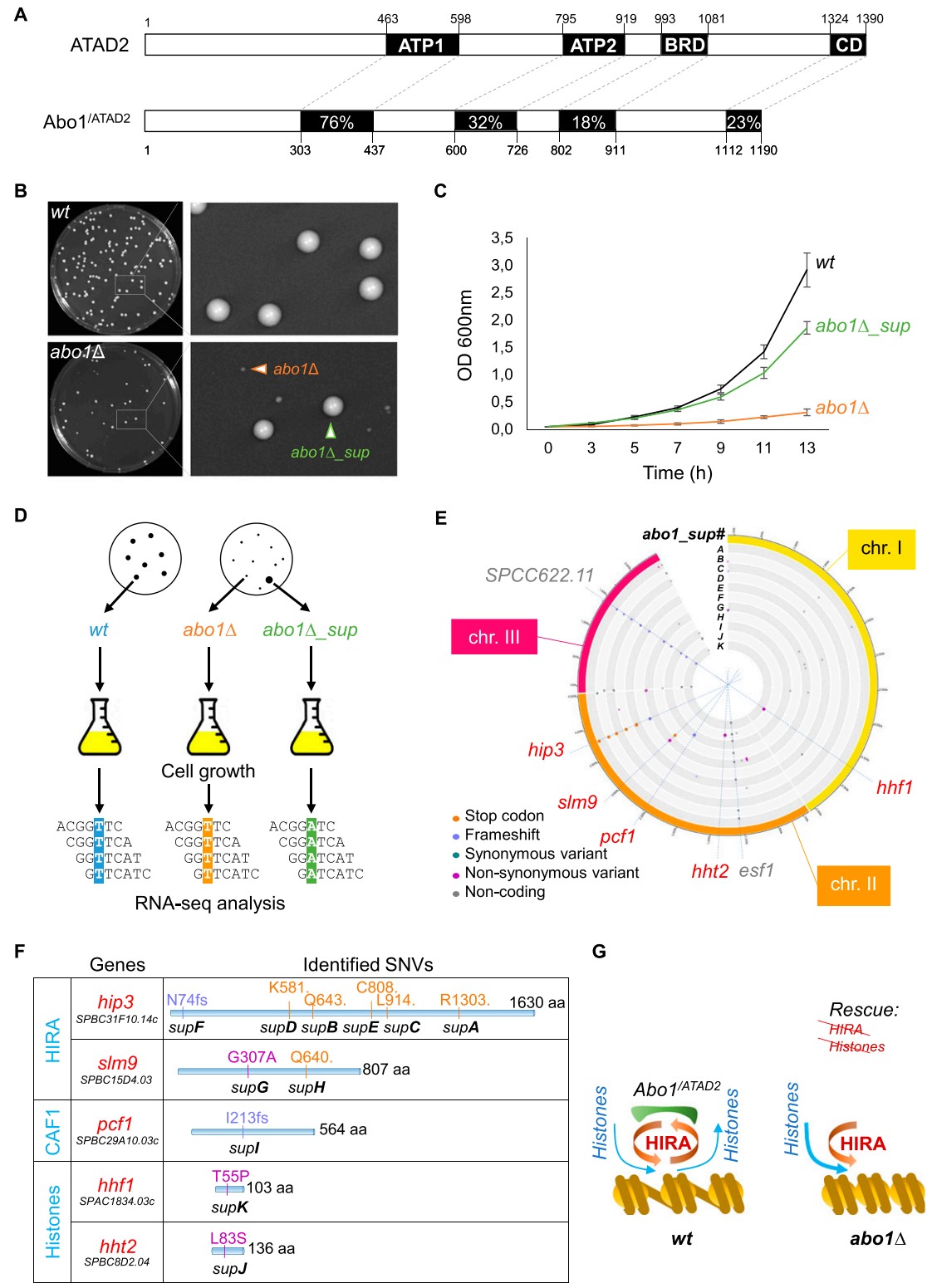

**Figure 1. In *S. pombe*, a deletion of the *abo1* gene causes a marked cell proliferation defect suppressed by mutations in HIRA or CAF1 complexes or in histones H3 or H4.**
**(A)** Schematic representation of human ATAD2 and *S. pombe* Abo1 proteins. The percentage of amino acid sequence identity is indicated for the most conserved domains. ATP1 and ATP2, AAA+ ATPase domain 1 and 2; BRD, Bromo domain; CD, C-terminal domain. **(B)** Photographs of wild-type (*wt*, SPV 8) and *abo1*Δ (SPV 3789) colonies grown on solid medium for 4 d, with magnified views to highlight the co-existence of small size colonies (orange arrowhead) and large size colonies (green arrowhead, which correspond to suppressor clones) on the *abo1*Δ plate. **(C)** Growth curves of *wt*, *abo1*Δ, and *abo1*Δ suppressor (*abo1*Δ_sup) cells obtained from three biological replicates

**Table 1. Genomic mutations linked to the suppression of *abo1*Δ cells growth defect.**

| Suppressor strains | Single-nucleotide variant | Amino acid modification | Protein name | Function (protein complex) |
|---|---|---|---|---|
| *abo1Δ_supA* | G>A (3778764) | R1303stop | | |
| *abo1Δ_supB* | G>A (3780744) | Q643stop | | |
| *abo1Δ_supC* | A>C (3779930) | L914stop | Hip3 | |
| *abo1Δ_supD* | T>A (3780930) | K581stop | | HIRA complex |
| *abo1Δ_supE* | A>T (3780247) | C808stop | | |
| *abo1Δ_supF* | ins T (3782449) | N74fs | | |
| *abo1Δ_supG* | G>T (3016457) | G307A | Slm9 | |
| *abo1Δ_supH* | C>T (3017455) | Q640stop | | |
| *abo1Δ_supI* | ins A (2539317) | I213fs | Pcf1 | CAF1 complex |
| *abo1Δ_supJ* | T>C (1365405) | L83S | Hht2 (H3) | Histones |
| *abo1Δ_supK* | T>G (4699234) | T55P | Hhf1 (H4) | |

Detailed description of the single-nucleotide variants in genes encoding subunits of HIRA and CAF1 histone chaperone complexes or histone H3 and H4 found in the 11 *abo1*Δ_sup (A to K) clones (also see Fig S3). The genomic positions of the single-nucleotide variants are indicated in brackets. >, substitution; ins, insertion; fs, frameshift; stop, non-sense.

histone overloading by enhanced histone chaperone activity is a source of cell toxicity.

## Atad2 controls HIRA and FACT chromatin interaction

The data obtained on the suppressors of the *abo1*Δ cell growth defect prompted us to test the occurrence of a functional interplay between Atad2 and HIRA in mammalian cells. We also included FACT because of the published data on the interaction of Atad2/Yta7/Abo1 with FACT (Lambert et al, 2010; Gal et al, 2016; Morozumi et al, 2016).

The analysis of HIRA and FACT content in cells lacking Atad2 showed a systematic accumulation of HIRA in all the tested cell systems. These included human hepatocellular carcinoma HepG2 and HeLa cells, as well as mouse ES cells (Fig 3A).

To define the origin of the accumulation of the HIRA histone chaperone, we also tested the levels of the mRNAs encoding FACT and HIRA. Fig 3A (right panels) shows that Atad2/ATAD2 has no role in the control of the levels of the considered mRNAs, suggesting that the accumulation of HIRA in the absence of ATAD2 results from protein stabilisation. One reason for the observed accumulation of HIRA could be its increased residence time on chromatin.

To test this hypothesis, we carried out a high-resolution genomic mapping of FACT and HIRA in the presence or absence of Atad2. We chose mouse ES cells, where Atad2 is highly expressed and which present a hyperdynamic chromatin (Meshorer et al, 2006; Morozumi

et al, 2016) to undertake this high-resolution analysis of the role of Atad2 in controlling FACT and HIRA interactions with chromatin. To this end, we designed a parallel ChIP-seq mapping of Ssrp1 and HIRA on MNase-digested chromatin in ES cells that were either wild-type or KO for *Atad2*. Sequencing and mapping of the immunoprecipitated DNA fragments shows that, whereas under our experimental conditions, we could not capture any significant amounts of FACT or HIRA on the gene transcriptional start sites (TSSs) in wild-type cells, an accumulation of HIRA and to some extent FACT, was observed on gene TSSs in the absence of Atad2 (Fig 3B).

## Atad2 controls histone deposition at gene TSSs by FACT and HIRA

Inspection of the nucleosome positioning around gene TSSs as a function of gene transcriptional activity shows the presence of the expected "nucleosome-free region" (NFR) within a region containing well-positioned nucleosomes, −1, +1, and +2 at the TSSs of active genes. Nucleosome positioning, which is very strong at +1 and +2, gradually fades at position +4/+5 (Fig 4A, input). In addition, the depth of the NFR region appears clearly proportional to the extent of gene transcriptional activity (Fig 4A, Inputs in WT and *Atad2* KO). In *Atad2* KO cells, the global nucleosomal organization around gene TSSs was not significantly altered. However, the mapping of HIRA- and Ssrp1-associated nucleosomes showed a very clear accumulation of both proteins on nucleosomes −1, +1, and +2 (Fig 4A and B,

done with three different clones (SPV 3,789, 3,790, and 3,791) and grown in liquid medium over the indicated times. **(D)** Diagram of the screen conducted to identify mutations responsible for the suppression of *abo1*Δ cell growth's defect by using RNA sequencing. Single colonies of *wt*, *abo1*Δ (with severe growth defect) and *abo1*Δ_sup cells were picked to inoculate liquid cultures and total RNA was purified. Massive parallel sequencing from the purified total RNAs depleted from rRNAs was used to detect SNVs (for details see Table S1). Spotting assays of wild-type, abo1Δ and abo1Δ_suppressor clones are shown in Fig S1. **(E)** Circos plot showing the SNVs identified in 11 independent isolates of *abo1*Δ_sup (*A* to *K*) along the three *S. pombe* chromosomes (chr. I, II and III). The positions of all SNVs identified, either in coding or non-coding regions of the genome, are highlighted by dots colored according to the nature of the mutations, as indicated. Blue dashed lines highlight SNVs localizing within the coding sequence (cds) of subunits of histone chaperone complexes (*hip3*, *slm9*, and *pcf1*) or histones (*hht2* and *hhf1*) genes, as well as two SNVs, in the cds of *SPCC622.11* or *esf1* genes, identified in a majority of *abo1*Δ_sup isolates but that do not contribute to the suppression of *abo1*Δ cells growth defect. Spotting assays of backcrossed *abo1*Δ_suppressor clones are shown in Fig S3. **(F)** Position and nature of the amino acid modifications induced by the SNVs found in the 11 *abo1*Δ_sup isolates. The color code of the mutations is the same as in (E). fs: frameshift, dot: non-sense mutation. **(G)** Scheme showing the functional interplay between Abo1 and HIRA complex allowing correct loading of histones on chromatin in wild-type cells (left part of the panel), and the potential effect of a lack of Abo1 on HIRA's function and histone loading (right part of the panel) in *abo1*Δ cells.

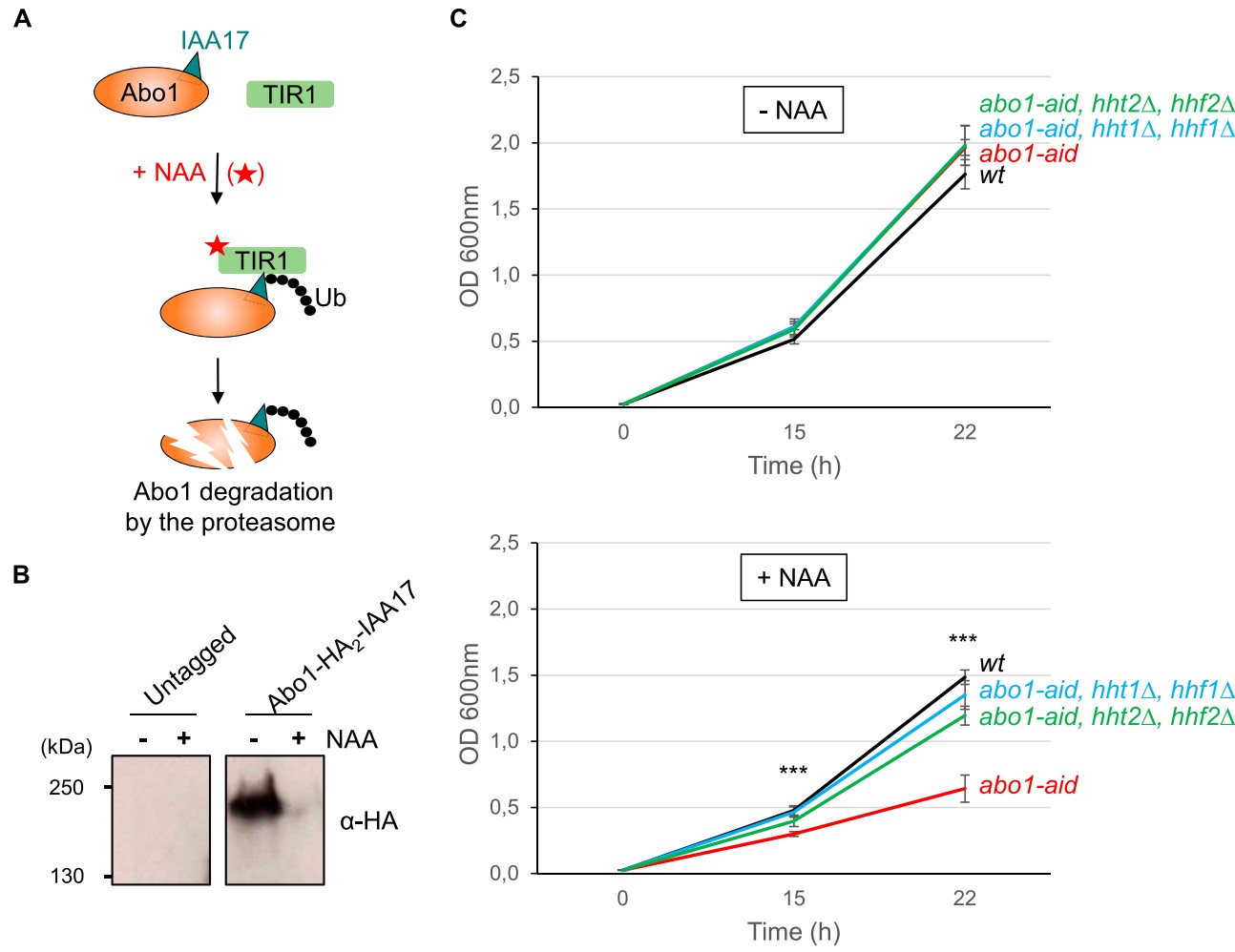

**Figure 2. A reduction of H3 and H4 genes copy number suppresses the growth defect caused by Abo1 depletion.**
**(A)** Scheme of the conditional knock-down strategy of *abo1* using the auxin-inducible degron system. TIR1 F-box proteins from *Arabidopsis thaliana* and *Oryza sativa* (rice) are expressed fused to *S. pombe* Skp1 (green rectangle TIR1), whereas endogenous Abo1 is fused to the 2xHA-IAA degron (blue triangle, IAA17) double tag. Addition of synthetic auxin 1-naphthaleneacetic acid (NAA) to the medium promotes association between Skp1–Cullin–F-box (SCF) E3 ubiquitin ligases complex and the degron tag, leading to subsequent ubiquitination of Abo1 and its degradation by the proteasome. **(B)** Western blot showing the level of Abo1-HA$_2$-IAA17 fusion protein, 6 h after addition of DMSO (–) or NAA (+) in cells expressing TIR1-Skp1 proteins (right panel, Abo1-HA$_2$-IAA17, SPV 4530). Protein extracts from untagged cells (left panel, untagged, SPV 4451) show the absence of cross-reaction with the $\alpha$-HA antibody. Molecular weight markers (kD) are shown on the left. Kinetics of the effect of the conditional knock-down of Abo1 on cell growth are shown in Fig S4. **(C)** Growth curves of *wt* (SPV 4,451), *abo1-aid* (SPV 4,530, 4,531, and 4,532), *abo1-aid, hht1Δ, hhf1Δ* (SPV 4,817, 4,818 and 4,819) and *abo1-aid, hht2Δ, hhf2Δ* (SPV 4,822, 4,823 and 4,824) cells obtained from three biological replicates issued from two different clones. Cells were grown in liquid medium containing DMSO (upper graph, – NAA) or 0.5 mM NAA (lower graph, + NAA) for the indicated times before measuring the OD at 600 nm. In the presence of NAA, *t* test indicates a significant difference in *abo1-aid* cells' growth, compared to all three other cell cultures, ***$P < 0.001$.

Ssrp1 and HIRA). This accumulation is proportional to gene transcriptional activity, meaning that the highest levels of FACT/HIRA-associated nucleosomes were observed on genes with the highest transcriptional activity (Fig 4A).

Interestingly, in *Atad2* KO cells, the Chip-seq mean read coverage profiles suggest that a structure with the characteristics of a nucleosome (180 bp of DNA from dyad to dyad) associated with HIRA and Ssrp1 appears over the NFR of highly expressed genes, which is not present on the TSS of genes with low activity (Fig 4A and B, red arrow). This particular observation suggests that at NFRs of active genes, both HIRA and FACT continuously deposit nucleosomes, but that these nucleosomes are highly unstable and disappear rapidly

thanks to Atad2 activity. In the absence of Atad2, the trapping of HIRA and FACT on these nucleosomes and their inefficient disassembly allows their detection.

Altogether, these results strongly suggest that, in the absence of Atad2, the residence time of HIRA and FACT on chromatin significantly increases. These observations in ES cells, together with our findings in *S. pombe* clearly suggest that Atad2 controls the dynamics of histone chaperones–chromatin interaction, especially that of HIRA. The absence of Atad2 would significantly decrease the dynamics of interaction between histone chaperones and chromatin and lead to a disrupted nucleosome assembly–disassembly equilibrium with a net increase in nucleosome assembly on these regions (Fig 4C).

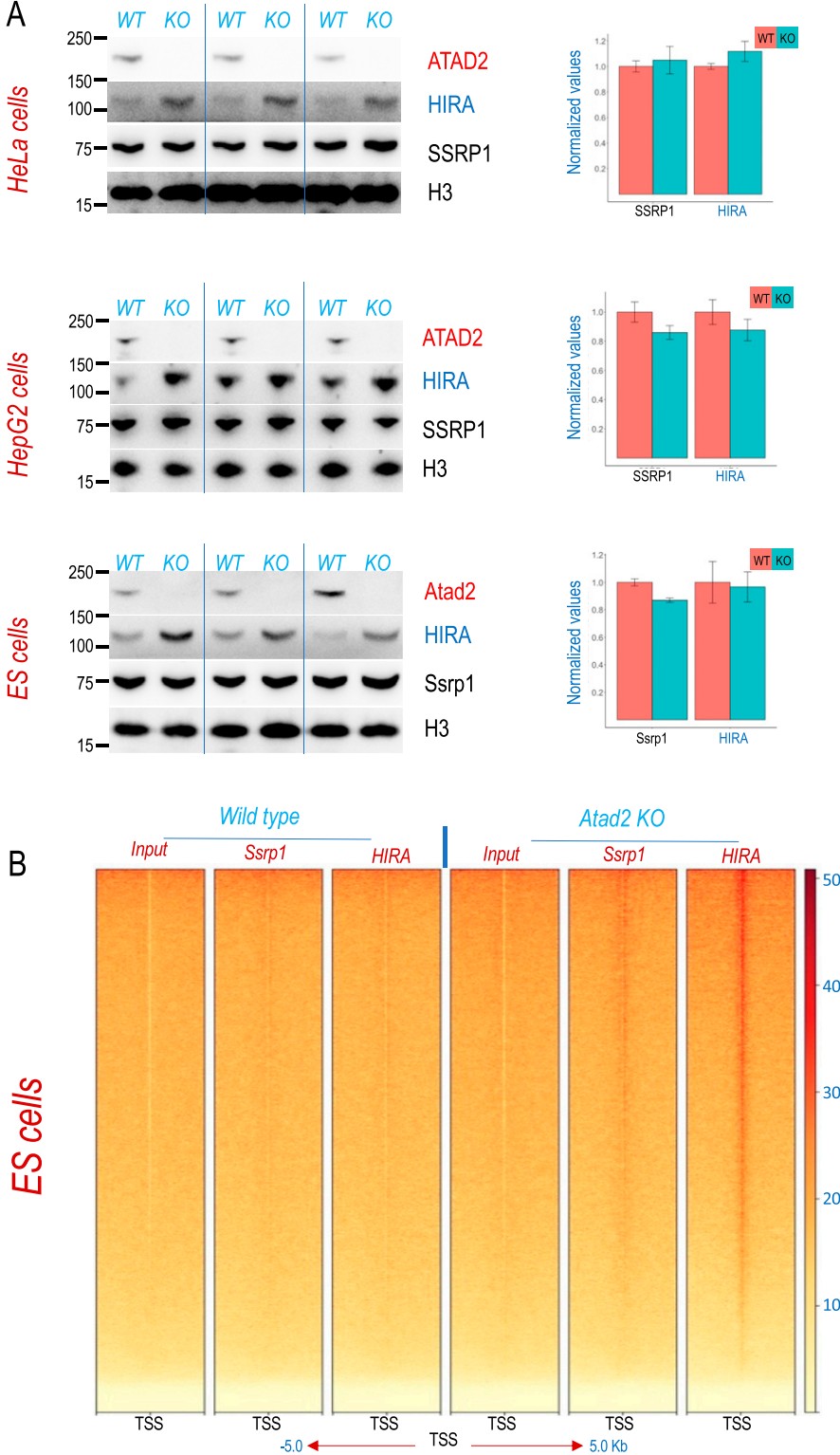

**Figure 3. Atad2 controls FACT and HIRA interaction with chromatin.**
**(A)** Extracts from human HeLa and HepG2 and mouse embryonic stem cells after *ATAD2* KO (by CRISPR/Cas9 system) were probed with the indicated antibodies. Three independent different biological replicates are shown (left panels). SSRP1- and HIRA-encoding mRNAs were also quantified from parallel cultures of the same cell lines (three independent biological replicates) by RT-qPCR. The mean values were calculated from triplicates for each biological replicate (n = 3) and are shown in the bar diagrams for wild-type (red) and ATAD2 KO (green) cells (right panels). Error bars indicate the standard errors of the mean values.
**(B)** Mononucleosomes generated after extensive digestion of chromatin from wild-type and *Atad2* KO embryonic stem cells were immunoprecipitated with the indicated antibodies and the immunoprecipitated DNA fragments were sequenced. A heat map of read density over −5 to +5 kb centred on all transcription start sites is shown (the density scale is shown on the right). Source data are available for this figure.

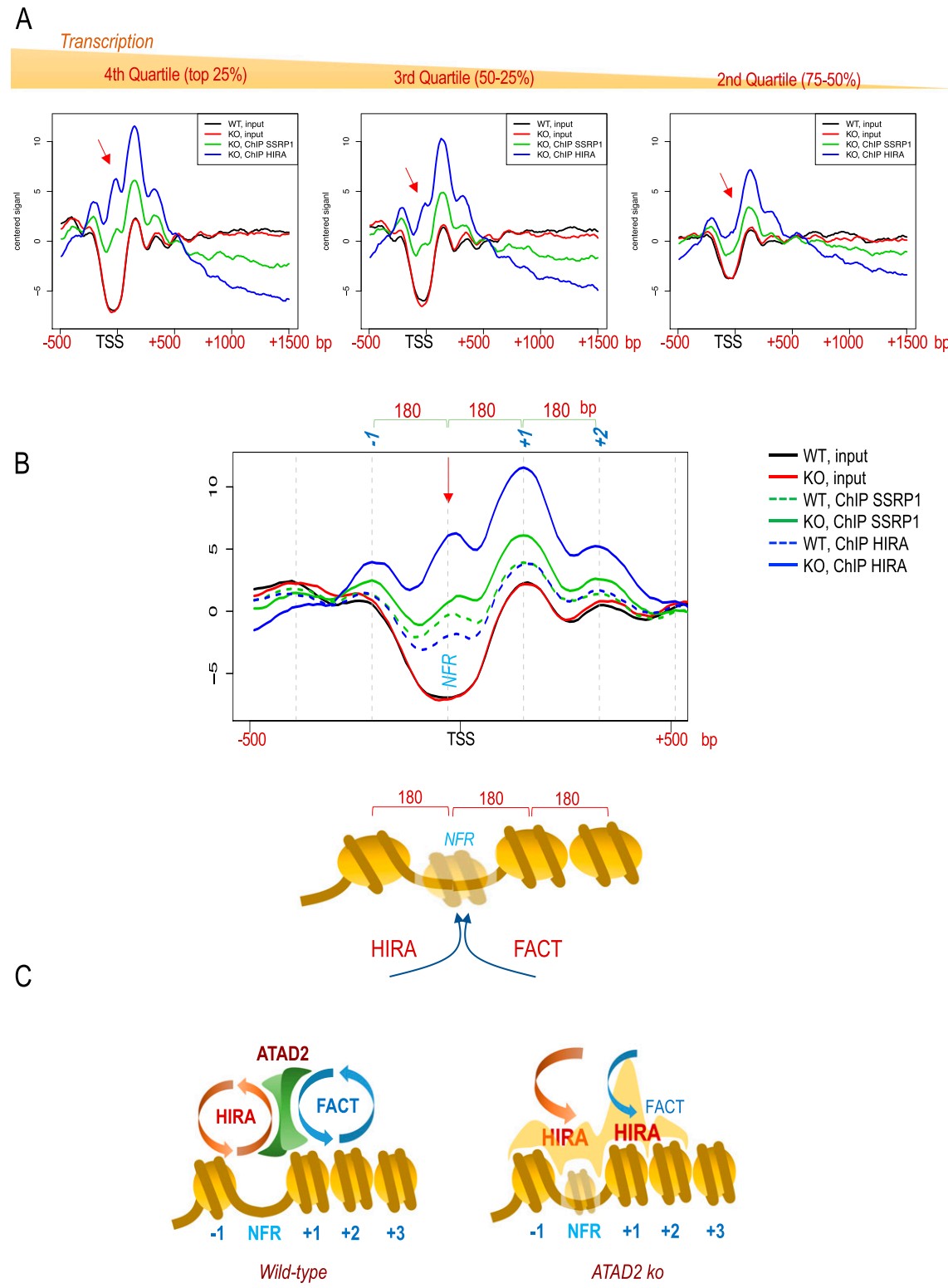

**Figure 4. Atad2-dependent control of histone deposition by FACT and HIRA in embryonic stem cells.**
**(A)** RPKM-normalized read coverage mean values over a region spanning from −500 to 1,500 bp with respect to the gene transcriptional start site (TSS) are shown for the input DNA (black line, wild type; red line, *Atad2* KO). DNA co-immunoprecipitated with HIRA (blue line) and Ssrp1 (green line) from *Atad2* KO embryonic stem cells are also shown over the input signal. For this analysis, the gene TSSs were grouped into quartiles as a function of gene transcriptional activity calculated from our own RNA-seq data (Morozumi et al, 2016), and the TSS corresponding to the top (fourth quartile) most expressed genes, third quartile (mid-expression) and second quartile (low expression) are shown separately from left to right. The red arrow indicates the peak of read coverage value present at the HIRA-bound nucleosome-free region (NFR)

# Discussion

Our parallel studies of ATAD2's function in distinct models, *S. pombe*, human cancer cells and mouse ES cells, consistently support that ATAD2 is a critical regulator of histone deposition by FACT and HIRA. Indeed, the results of the functional studies in *S. pombe* highlight the ability of Abo1 to control the activity of the histone chaperone complex, HIRA. The sequencing of 11 independent *S. pombe* suppressor *abo1*Δ clones identified unrelated inactivating mutations in two HIRA complex subunits in eight of them. These results support the hypothesis that, in the absence of Abo1, HIRA activity becomes toxic to the cells. Considering the other two suppressor mutations, affecting H3 and H4 dosage as well as the CAF1 histone chaperone, our observations suggest that the deregulated activity of HIRA and probably to some extent of CAF1, could lead to a trapping of these histone chaperones on chromatin and/or histone overloading.

The rescue of the Abo1 depletion phenotype by a targeted histone underdosage further supports this hypothesis. Indeed, the effects of an inactivation of these chaperones and of histone underdosage suggest that histone dependent-trapping of histone chaperones could be at the origin of cell toxicity.

In another unrelated model, mouse ES cells, the entrapment of FACT and HIRA on chromatin in the absence of Atad2 was directly visualized. Indeed, ChIP-seq mapping of FACT and HIRA revealed that, in cells lacking Atad2, the binding of both HIRA and FACT to nucleosomes surrounding the NFR of active genes was increased. In the absence of Atad2, HIRA and FACT accumulated on the −1, +1, and +2 nucleosomes, as well as on the normally nucleosome free regions. This important observation also helps to better understand the true nature of NFRs at actively transcribed gene TSSs. According to these data, we can propose that the NFR is not a genuinely "nucleosome-free" region but contains a nucleosome which is permanently oscillating between assembly and disassembly with the latter state being reduced in ATAD2 deficient cells. FACT and HIRA would favour the assembly of this nucleosome whereas Atad2 could favour its disassembly.

These data are also in agreement with the previously published mapping of HIRA on the genome of HeLa and ES cells, which identified active gene TSSs close to the NFR as prominent sites of HIRA chromatin interaction (Banaszynski et al, 2013; Pchelintsev et al, 2013). It is of note that in our hands, this specific binding is particularly visible in *Atad2* KO ES cells, certainly because of an enhanced interaction of HIRA and FACT with chromatin in the absence of Atad2. Finally, our conclusions on the role of ATAD2 and Abo1 perfectly agree with results previously reported in *S. cerevisiae* expressing an ortholog of ATAD2, known as Yta7 (Cattaneo et al, 2014). Functional studies of Yta7 revealed that the protein's major activity is to oppose the activity of Rtt106, a H3/H4 deposition chaperone (Lombardi et al, 2015). These authors clearly demonstrated that in the absence of Yta7, Rtt106-dependent histone deposition is enhanced, leading to an increased nucleosome density.

Overall, these studies also generalize the idea of a need for ATP hydrolysis and the involvement of ATPase activity in the control of histone chaperone activity. Along these lines, a recent publication demonstrated that DNAJC9 recruits HSP70 ATPase activity into the histone supply chain (Hammond et al, 2021). The authors also showed that histone-bound DNAJC9 bearing a mutation that disabled the protein from mediating ATP hydrolysis by HSP70 became trapped on chromatin. The situation is similar to that of HIRA and FACT which, in the absence of ATAD2 are also trapped on chromatin, especially at the TSS of active genes. We can therefore propose that ATAD2, similar to DNAJC9-HSP70, brings the necessary energy to histone chaperones to ensure the dynamic histone deposition–removal cycle (Hammond et al, 2017; Mendiratta et al, 2019). In addition, this recent discovery of the involvement of DNAJC9-HSP70 in the histone supply pathway suggests that the histone chaperone-associated ATPase could be multiple and that DNAJC9-HSP70 recruitment could to some extent compensate for the lack of ATAD2.

This work not only highlights the long-awaited role of ATAD2, which had remained obscure for many years despite intense investigations but also reveals an essential level in the regulation of FACT-HIRA chromatin interaction.

# Materials and Methods

### Cell lines and cell cultures

Human Hepatocyte carcinoma cell line HepG2, and Human endocervical adenocarcinoma cell line HeLa were obtained from National Collection of Authenticated Cell. Mouse ES cell 46C was a gift from Philippe Avner. The gap-repair recombineering technique was used in ES cells to generate 46C^Atad2−Tag cell line as described previously (Morozumi et al, 2016). HepG2 and HeLa and the derivative cells were cultured in DMEM (Low glucose, 1 g/l, Gibco) supplemented with 10% FBS, 4 mM L-glutamine, and 1% penicillin–streptomycin. 46C ESCs and the derivative cells were cultured in DMEM medium containing LIF (ESG1107; Millipore), non-essential amino acids, 1% penicillin–streptomycin, $\beta$-mercaptoethanol, and supplemented with 15% foetal bovine serum (Invitrogen), on 0.1% gelatin-coated tissue culture dishes. All cells were incubated at 37°C with 5% $CO_2$.

### CRISPR/Cas9–mediated KO cell line

HepG2 and HeLa cells were co-transfected with lenti-Cas9 and lenti-sgRNAs plasmids (targeting sequences: 1. AATCTTAATATGTACACAAG;

---

region of highly active genes, which disappears on the NFRs of the less active gene TSSs. **(B)** Input and ChIP read signals shown in panel (A) for the top 25% active genes are shown at higher resolution to visualize the distance separating neighbouring nucleosomes from dyad to dyad. A schematic representation of the nucleosomal organization over gene TSSs is shown below. **(C)** Models summarizing the ChIP-seq mapping data of nucleosome distribution and HIRA and FACT localization in wild-type and *Atad2* KO active gene TSSs are shown. In wild-type cells, Atad2 ensures a dynamic interaction of HIRA and FACT with chromatin at the gene TSSs, maintaining an equilibrium between histone deposition and removal. In *Atad2* KO cells the residence time of HIRA and FACT on nucleosomes is significantly increased, especially on the NFR region.

2. GAAACAACTGATAATCAAGA; 3. TAGGCAGTTGGCCAGACAGC) (Shalem et al, 2014). 2 d after transfection, resistant cells were selected using puromycin (1.0 µg/ml; Sigma-Aldrich) and blasticidin (10.0 µg/ml; Sigma-Aldrich). After 2 d, resistant cells were recovered in normal medium and cultured for another 2 d. After being seeded into 96-well plates for 2–3 wk, single cell clones were obtained and genotypes of each allele of single cell clones were analysed through Sanger sequencing. KO clones were further confirmed by Western blotting. An *Atad2* KO 46C[Atad2-Tag] ES cell line (Morozumi et al, 2016) was also generated as above, with the following targeting sequence: ATCGACGGCGTCCAAGGCGG.

## RT-qPCR

Cell lines were lysed in 500 µl TRIzol reagent (Invitrogen). Total RNA was obtained using standard procedure. cDNA was produced from 2.5 µg RNA with AffinityScript Multi Temperature cDNA Synthesis Kit (Agilent Technologies) using random hexamers according to the manufacturer's instructions. Quantitative PCR (qPCR) was performed with Brilliant III Ultra-Fast SYBR Green QPCR Master Mix (Agilent Technologies) on Mx3005p (Stratagene) using primers as listed in Table S2.

## Protein extraction and immunoblotting

Proteins were extracted in urea and sonicated and mixed with Laemmli sample buffer. Immunoblotting analyses were carried out according to the standard procedures using the antibodies listed in Table S3. Revelation was performed with Vilber Chemiluminescence system (Vilber).

## *S. pombe* strains and genetics

Genotypes of strains used in this study are listed in Table S4. *S. pombe* cells were cultured at 30°C in liquid or solid rich medium except otherwise mentioned. Mutant *abo1Δ* cells were obtained using the PCR-based gene targeting method (Bahler et al, 1998) and by replacing *abo1* coding sequence with nourseothricin (NAT) selection gene. Positive transformants were obtained by growth on rich medium with 200 µg/ml of nourseothricin and validated by genomic PCR. The *abo1Δ_suppressor* cells analysed in this study appeared spontaneously after growth of *abo1Δ* cells in liquid cultures and individual colonies were isolated on solid rich medium. Unwanted SNVs present in the genomes of *abo1Δ_suppressor* clones were removed by backcrossing the cells with *wt* cells. Backcrosses were done by plating cells on SPAS. Spores issued from the backcrosses were isolated under microscope and germinated on solid rich medium. Screening for the presence of mutated allele of *hip3* (in *abo1Δ_supA* and *abo1Δ_supC* isolates) or *slm9* (in *abo1Δ_supH*) genes, and of *wt* alleles in *SPCC622.11* and *esf1* genes, was verified by PCR amplification of the corresponding genomic regions (Biomixred from Bioline) followed by DNA sequencing (Eurofins). Primers used for genotyping are listed in Table S5.

## Protein knock-down using the auxin-inducible degron (AID) system

The insertion of the *2xHA-aid* tag at the 3′ end of *abo1* CDS (construct called *abo1-aid*) was performed in cells expressing *S. pombe* Skp1 in fusion to the *A thaliana* and *Oryza sativa* TIR1 F-box proteins (Egan et al,

2014). To ensure an efficient degron-dependent knock-down of the Abo1-2xHA-AID protein, all cultures were performed at 25°C, in YEA (Fig 2C) or EMMC (Figs 2B and S4) (MP Biomedicals) medium. After growing cells to saturation, cell cultures were diluted to an OD600 of 0.025–0.1 in fresh medium containing 1-naphthaleneacetic acid (from Sigma-Aldrich) to a final concentration of 0.5 mM. Control cultures were treated with an identical volume of DMSO (NAA vehicle). Western blot with an antibody against anti-HA tag (Ab9110) was performed on total protein extract from cells cultivated for 6 h in the presence of NAA to verify the degradation of Abo1-2xHA-AID protein.

## RNA-seq, sequence alignment, and variant calling

8 µg of total RNA treated with Dnase I (Roche) and purified by acid phenol and chloroform extractions from 25 ml of log phase *S. pombe* cell cultures were sent for high-throughput sequencing. RNA quantity was determined using the Qubit Fluorometer and RNA integrity validated using the Fragment Analyzer system with the PROSize 3.0 software (Agilent Technologies). The RQN of each sample were ranging from 8.8 to 10. Total RNA samples were subjected to ribozero depletion (Illumina Ribo-Zero Plus kit). Library construction was carried out according to the "long-non-coding libraries" protocol (Novogene). Briefly, RNA was fragmented randomly prior the synthesis of cDNA first strand and U-contained second strand. After purification by AMPure XP beads, terminal repair and polyadenylation, sequencing adapters were ligated to cDNAs ends. After size selection and degradation of second strand by the USER enzyme, the strand-specific cDNA library was generated after the final PCR enrichment. Massively parallel sequencings were performed using either Illumina HiSeq2000 (single end, strand-specific sequencing, read length of 50 nt and 10 million reads per sample) or Illumina NovaSeq (paired-end, strand-specific sequencing, reads length of 150 nt, and 50 million reads per sample). Quality of the raw sequencing reads was assessed using FastQC (www.bioinformatics.babraham.ac.uk/projects/fastqc/) and MultiQC. The reads (FASTQ) were processed and analysed with the Lasergene Genomics Suite version 15 (DNASTAR) using default parameters. Paired-end reads were uploaded onto the SeqMan Ngen (version 15, DNASTAR) platform for reference-based assembly and variant calling using the *Schizosaccharomyces_pombe* genome (pombase.org) as the reference template. Variant calls were filtered with the following criteria: Q-call ≥ 60, variant depth ≥ 50, and absence in the parental wild-type strains (Table S1). Mutations were plotted on a Circos plot using Circa (OMGenomics.com).

## ChIP-seq

ChIP assays for SSRP1 and HIRA were carried out as previously described (Barral et al, 2017; Shiota et al, 2018; Gao et al, 2021). Five X 10⁷ 46C[Atad2-Tag] WT and *Atad2* KO cells were pelleted, washed twice with PBS (with 10 mM Sodium Butyrate), and lysed in 1.5 ml lysis buffer (15 mM Tris–HCl, pH 7.4, 60 mM KCl, 15 mM NaCl, 0.34 M sucrose, 2 mM EDTA, 0.5 mM EGTA, 0.65 mM spermidine, 1 mM DTT, 0.05% Triton-X 100, 1% glycerol, 1X protease cocktail inhibitors, 1X Phosphatase Inhibitor cocktail 1, and 10 mM sodium butyrate) and incubated for 5 min at 4°C. Cell nuclei were pelleted by centrifugation at 200*g* for 15 min at 4°C and resuspended in wash buffer

(15 mM Tris–HCl, pH 7.4, 60 mM KCl, 15 mM NaCl, 0.34 M sucrose, 0.65 mM spermidine, 1 mM DTT, 1× protease cocktail inhibitors, 1× Phosphatase Inhibitor cocktail 1, and 10 mM sodium butyrate) and centrifuged again. Cell nuclei were resuspended in MNase buffer (10 mM Tris–HCl, pH 7.5, 10 mM KCl, 2 mM $CaCl_2$, 1 mM DTT, 1× protease cocktail inhibitors, 1 × phosphatase inhibitor cocktail 1, and 10 mM sodium butyrate). Then the cell nuclei solution was digested by micrococcal nuclease S7 (5U MNase per 100 μg nuclei; Sigma-Aldrich) at 37°C for 7 min to obtain mononucleosomes. Small aliquots of mononucleosome solutions were collected for input and used to check the efficiency of digestion before immunoprecipitation (IP).

IPs were carried out as follows: 5 μg anti-Ssrp1 and anti-HIRA antibodies were coupled with 50 μl Dynabeads and protein G (Thermo Fisher Scientific), respectively, according to the manufacturer's instructions. Digested mononucleosome solutions were diluted with LSDB500 buffer (50 mM Hepes, pH 7.0, 3 mM $MgCl_2$, 500 mM KCl, 20% glycerol, 1X protease cocktail inhibitors, 1× phosphatase inhibitor cocktail 1, and 10 mM sodium butyrate) to achieve the final KCl concentration of 350 mM. For each reaction, 100 μg of chromatin was incubated with antibody-coupled beads at 4°C for 16 h. Immunoprecipitated beads were then washed three times with LSDB350 buffer (50 mM Hepes, pH 7.0, 3 mM $MgCl_2$, 350 mM KCl, 20% glycerol, 1 × protease cocktail inhibitors (Complete; Roche), 1× phosphatase inhibitor cocktail 1, and 10 mM sodium butyrate) and once with elution buffer (10 mM Tris–HCl, pH 8.5, 1 mM EDTA, and 10 mM sodium butyrate). ChIP samples were eluted from beads with elution buffer containing 1% SDS at 65°C for 20 min and were purified by phenol–chloroform extraction and ethanol precipitation in parallel with input samples.

For sequencing, ChIP libraries were prepared using MicroPlex Library Preparation Kit v2 (Diagenode) according to the manufacturer's instructions. Each library was quantified on Qubit with Qubit dsDNA HS Assay Kit (Life Technologies) and size distribution was examined on the Fragment Analyzer with High Sensitivity NGS Fragment Analysis kit (Agilent). Libraries were then sequenced on a high-output flow cell (400 M clusters) using the NextSeq 500 High Output v2.5 150 cycles kit (Illumina), in paired-end 75/75 nt mode, according to the manufacturer's instructions at the TGML Platform of Aix-Marseille Université.

Raw fastq files were processed by 5 prime trimming, keeping 30-bp-length fragments, using "fastx_trimmer" (with options -l 30 -Q33). The trimmed fastq files were aligned on the UCSC mm38 genome using the Bowtie2 with options −end-to-end, −no-mixed, −no-discordant.

The aligned read counts were converted into a 10-bp bin matrix of the signal 2 Kb upstream and downstream genes TSS, using computeMatrix (from the package deepTools2, Ramirez et al, 2016), heat maps and profiles were generated using the respective deepTools2 packages, plotHeatmap, and plotProfile.

The bam signal was normalized and smoothed using bamCoverage with options: −binSize 4 −minMappingQuality 30 −normalizeUsing RPKM.

To constitute groups of genes according to their expression levels in ES cells, we used RNAseq data from control ES cells in GSE70314 (GSM1723639 et GSM1723640). Protein-coding genes from NCBI Reference Sequence Database (RefSeq) were selected and grouped by quartile of absolute RPKM expression in si-scramble samples from GSE70314. These groups were exported as .bed files.

Heat maps were produced from BigWig files and quartile of expression .bed files using computeMatrix and plotHeatmap with options:

reference point −referencePoint TSS −binSize 10 −beforeRegionStartLength 500 −afterRegionStartLength 1500 −sortRegions descend. TSS profiles were generated from computeMatrix outputs with a custom R script.

# Data Availability

The data generated for this work corresponding to the ChIP-seq experiments with HIRA and SSRP1 antibodies in wild-type (WT) and ATAD2 KO (KO) ES cells are available on GEO (accession number GSE178647).

# Supplementary Information

# Acknowledgements

This research benefited from Plan Cancer Pitcher (S Khochbin) and from Merck Sharpe and Dohme (MSD) Avenir ERICAN (S Khochbin and A Verdel) programs. S Khochbin and C Petosa laboratories are also supported by Agence Nationale de la Recherche (ANR) Episperm4 program and by the "Université Grenoble Alpes" ANR-15-IDEX-02 LIFE (S Khochbin) and SYMER (S Khochbin and C Petosa) programs. Additional supports were from the Foundation Association pour la Recherche Contre le Cancer (ARC) (PGA program No RF20190208471, coordinated by S Khochbin and D Puthier), from the French National Cancer Institute (INCa), and the French Institute for Public Health Research (IreSP) (INCa_13641), as well as from the Cancer ITMO (Multi-Organisation Thematic Institute) of the French Alliance for Life Sciences and Health (AVIESAN) "Contribution of Mathematics and Informatics to Oncology" (MIC) program. High-throughput sequencing was performed at the Transcriptomique et Genomique Marseille-Luminy (TGML) Platform, supported by grants from Inserm, Groupe d'Interet Scientifique (GIS) IBiSA, Aix-Marseille Université, and ANR-10-INBS-0009-10. A Verdel laboratory is supported by ANR "RNAgermSilence" and "HeatEpiRNA" programs.

## Author Contributions

T Wang: investigation.
D Perazza: investigation.
F Boussouar: investigation.
M Cattaneo: investigation.
A Bougdour: formal analysis.
F Chuffart: formal analysis and investigation.
S Barral: investigation.
A Vargas: investigation.
A Liakopoulou: formal analysis and investigation.
D Puthier: formal analysis and investigation.
L Bargier: investigation.
Y Morozumi: investigation.
M Jamshidikia: investigation.
I Garcia-Saez: investigation.
C Petosa: formal analysis, investigation, and writing—review and editing.
S Rousseaux: conceptualization, formal analysis, supervision, and writing—original draft, review, and editing.

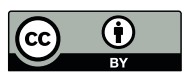

A Verdel: conceptualization, supervision, funding acquisition, project administration, and writing—original draft, review, and editing.

S Khochbin: conceptualization, formal analysis, supervision, funding acquisition, project administration, and writing—original draft, review, and editing.

## Conflict of Interest Statement

The authors declare that they have no conflict of interest.

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
