## [Reviewer comments · Life Science Alliance]

Life Science Alliance

ATAD2 controls chromatin-bound HIRA turnover

Tao Wang, Daniel Perazza, Faycal Boussouar, Matteo Cattaneo, Alexandre Bougdour, Florent Chuffart, Sophie Barral, Alexandra Vargas, Ariadni Liakopoulou, Denis Puthier, Lisa Bargier, Yuichi Morozumi, Mahya Jamshidikia, Isabel Garcia-Saez, Carlo Petosa, Sophie Rousseaux, Andre Verdel, and Saadi Khochbin

DOI: <https://doi.org/10.26508/lsa.202101151>

Corresponding author(s): Saadi Khochbin, Inserm U1209 and Andre Verdel, CNRS, INSERM and University of Grenoble

Review Timeline:

Submission Date:	2021-07-07
Editorial Decision:	2021-07-12
Revision Received:	2021-08-26
Editorial Decision:	2021-09-13
Revision Received:	2021-09-14
Accepted:	2021-09-15

Scientific Editor: Novella Guidi

Transaction Report:

Please note that the manuscript was previously reviewed at another journal and the reports were taken into account in the decision-making process at Life Science Alliance. Since the original reviews are not subject to Life Science Alliance's transparent review process policy, the reports and author response cannot be published.

July 12, 2021

Re: Life Science Alliance manuscript #LSA-2021-01151-T

Saadi Khochbin
Institut Albert Bonniot
CNRS/INSERM, Université Grenoble-Alpes
Domaine de la Merci
La Tronche 38706 cedex
France

Dear Dr. Khochbin,

Thank you for submitting your manuscript entitled "ATAD2 maintains the equilibrium between nucleosome assembly and disassembly" to Life Science Alliance. The manuscript was previously assessed by expert reviewers in other journal and based on their comments we would like to encourage further consideration of this manuscript at LSA pending the revisions.

We understand that such a revision might need to be re-reviewed, in which case, I will walk the Reviewers through our transfer process.

Thank you for this interesting contribution to Life Science Alliance. We are looking forward to receiving your revised manuscript.

Sincerely,

- A letter addressing the reviewers' comments point by point.
- An editable version of the final text (.DOC or .DOCX) is needed for copyediting (no PDFs).
- High-resolution figure, supplementary figure and video files uploaded as individual files: See our detailed guidelines for preparing your production-ready images, <https://www.life-science-alliance.org/authors>
- Summary blurb (enter in submission system): A short text summarizing in a single sentence the study (max. 200 characters including spaces). This text is used in conjunction with the titles of papers, hence should be informative and complementary to the title and running title. It should describe the context and significance of the findings for a general readership; it should be written in the present tense and refer to the work in the third person. Author names should not be mentioned.

B. MANUSCRIPT ORGANIZATION AND FORMATTING:

September 13, 2021

RE: Life Science Alliance Manuscript #LSA-2021-01151-TR

Dr. Saadi Khochbin
CNRS, INSERM and University of Grenoble
Domaine de la Merci
La Tronche 38706 cedex
France

Dear Dr. Khochbin,

Thank you for submitting your revised manuscript entitled "ATAD2 controls chromatin-bound HIRA turnover". We would be happy to publish your paper in Life Science Alliance pending final revisions necessary to meet our formatting guidelines.

- please upload both your main and supplementary figures as single files
- please add ORCID ID for secondary corresponding) author-they should have received instructions on how to do so
- please add your main, supplementary figure, and table legends to the main manuscript text after the references section
- please revise call-outs in the manuscript text for Figures 2C (which is missing) and 3C
- please indicate the molecular weight alongside blots in figure 3A

LSA now encourages authors to provide a 30-60 second video where the study is briefly explained. We will use these videos on social media to promote the published paper and the presenting author (for examples, see <https://twitter.com/LSAjournal>). Corresponding or first-authors are welcome to submit the video. Please submit only one video per manuscript. The video can be emailed to contact@life-science-alliance.org

A. FINAL FILES:

B. MANUSCRIPT ORGANIZATION AND FORMATTING:

Sincerely,

Novella Guidi, PhD
Scientific Editor

Reviewer #1 (Comments to the Authors (Required)):

The authors have removed a large part of the data to present a clear message of the paper avoiding exaggerated interpretation of the obtained results. I strongly feel that the paper is now well-organized in comparison with the previous version. Thus I will support the acceptance of the paper in its current form in LSA.

September 15, 2021

RE: Life Science Alliance Manuscript #LSA-2021-01151-TRR

Dr. Saadi Khochbin
Inserm U1209
INSERM U1209, University Grenoble Alpes
Domaine de la Merci
La Tronche 38706 cedex
France

Dear Dr. Khochbin,

Thank you for submitting your Research Article entitled "ATAD2 controls chromatin-bound HIRA turnover". It is a pleasure to let you know that your manuscript is now accepted for publication in Life Science Alliance. Congratulations on this interesting work.

DISTRIBUTION OF MATERIALS:

Again, congratulations on a very nice paper. I hope you found the review process to be constructive and are pleased with how the manuscript was handled editorially. We look forward to future exciting submissions from your lab.

Sincerely,
